# Characteristics of the Complete Chloroplast Genome of *Pourthiaea* (Rosaceae) and Its Comparative Analysis

Zhengying Cao [1,2], Wenzhi Zhao [1,2], Yaxuan Xin [1,2], Weixiang Shen [1,2], Fei Wang [1,2], Qishao Li [1,2], Yuxiang Tu [1,2], Haorong Zhang [1,2], Zhanghong Dong [1,2] and Peiyao Xin [1,2,*]

1   Southwest Research Center for Engineering Technology of Landscape Architecture National Forestry and Grassland Administration, Southwest Forestry University, Kunming 650224, China
2   Key Laboratory of National Forestry and Grassland Administration on Biodiversity Conservation in Southwest China, Southwest Forestry University, Kunming 650224, China
*   Correspondence: xpytgyx@163.com

**Abstract:** *Pourthiaea* is of great ornamental value because it produces white flowers in spring and summer, red fruit in autumn, and their fruit does not fall in winter. In order to explore the genetic structure and evolutionary characteristics of the chloroplast genome of *Pourthiaea*, comparative genomics analysis and phylogenetic analysis were conducted using ten published chloroplast genomes of *Pourthiaea* from the NCBI database. The results showed that the chloroplast genomes of the ten species of *Pourthiaea* showed typical circular tetrad structures, and the genome sizes were all within the range of 160,159–160,401 bp, in which the large single copy was 88,047–88,359 bp, the small single copy was 19,234–19,338 bp, and the lengths of a pair of inverted repeats were 26,341–26,401 bp. The GC contents ranged from 36.5% to 36.6%. A total of 1017 SSR loci were identified from the chloroplast genomes of the ten species of *Pourthiaea*, including six types of nucleotide repeats. The gene types and gene distribution of the IR boundary regions of the chloroplast genomes of different species of *Pourthiaea* were highly conservative, with little variation. Through the sequence alignment of chloroplast genomes, it was found that the chloroplast genomes of the ten species of *Pourthiaea* were generally highly conservative. The variation mainly occurred in the spacer regions of adjacent genes. Through nucleic acid diversity analysis, three hypervariable regions were screened at Pi > 0.006, namely *trnQ(UUC)-psbk-psbl*, *accD-psal*, and *ndhF-rpl32-trnL (UAG)*. Phylogenetic analysis showed that the ten species of the genus *Pourthiaea* were clustered in the same branch and formed sister groups with the genus *Stranvaesia*, and that the support rate for the monophyly of the genus *Pourthiaea* was high. This study can serve as a reference for the breeding, genetic evolution, and phylogeny of *Pourthiaea*.

**Keywords:** *Pourthiaea*; chloroplast genome; sequence characteristic; phylogenetic relationship

## 1. Introduction

*Pourthiaea* refers to deciduous trees or shrubs of Maleae in Rosaceae. The native area of this genus includes the Eastern Himalayas to East Asia and IndoChina, mainly distributed across East Asia, South Asia, and Southeast Asia [1]. The *Pourthiaea* genus was established by Decaisne in 1874 and is noticeably different from other genera because of its deciduous leaves, pedicels and peduncles with nodular protrusions, and fruit flesh with stone cells [2]. The leaves of this genus are papery, and the flowers are mostly in umbels, corymbos, or compound corymbos [3]. White flowers bloom in spring and summer, while the fruit appears red in autumn and does not fall in winter [2]. Therefore, *Pourthiaea* has high ornamental value and is usually cultivated as an ornamental plant.

Rosaceae is the most widely distributed family of angiosperms, with 91 genera and approximately 4828 species [4]. It includes three subfamilies: Dryadoideae, Rosoideae, and Amygdaloideae [5,6]. In the Amygdaloideae, the intergeneric relationship of Maleae has

attracted much attention [7–9]. The genus *Pourthiaea*, which was investigated in this study, is one of the most controversial genera of Maleae. The relationship between *Pourthiaea* and its related genera has always been complex, especially with regard to *Photinia*. In 1820, Lindley established *Photinia* [10]. Since then, three evergreen species and two deciduous species originally belonging to *Crataegus* have been moved into the genus [11,12]. The species under *Photinia* include two types: evergreen and deciduous. Until 1874, Decaisne believed that the deciduous species of *Photinia* were obviously different from those of other genera in that they had tuberous protrusions on the pedicels and peduncles, and stone cells in the flesh, and thus established *Pourthiaea* [13]. In the past, some botanists incorporated the genus *Pourthiaea* into *Photinia* based on morphological data [14–17], while some botanists believe that it should be an independent genus [18–20]. With the development of molecular biology, molecular systematics based on molecular biology has provided a new theoretical basis for taxonomy. In 2011, Guo et al. [21] first used two chloroplast gene fragments (*trnL-trnF* and *psbA-trnH*) and one nuclear gene fragment (*nrITS*) to analyze the phylogenies of *Photinia*, *Pourthiaea*, *Stranvaesia*, and other related genera. The results showed that *Pourthiaea* formed a branch line with high support and did not share a branch with *Photinia*, which provides solid evidence for the independence of *Pourthiaea*. Since then, many scholars have used different nuclear gene fragments, chloroplast gene fragments, whole chloroplast genomes (cpDNA), or ribosomal DNA (nrDNA) to study the phylogenetic relationship between *Pourthiaea* and its relatives [6,8,22,23], and the results have shown that *Pourthiaea* and *Photinia* are located in different branches. Not only the relationship between genera but also the division of species within the genus *Pourthiaea* is controversial, and the phenomenon of synonyms is serious. Liu et al. [2] revised the classification of the *P. villosa* complex, in which only one species, *P. villosa,* was identified, but it included 56 synonyms. Then, they classified the *P. blinii* complex, *P. salicifolia* complex, *P. sorbifolia* complex, and *P. arguta* complex [24] and finally provided a taxonomic list of *Pourthiaea* [25] for further taxonomic and evolutionary research.

Chloroplasts widely exist in eukaryotes and are an important site for photosynthesis and other metabolic processes. As semiautonomous organelles, they have a complete genome [26]. The chloroplast genomes of most higher plants each have a double-stranded circular tetrad structure, including a large single copy (LSC), a small single copy (SSC), and two inverted repeat sequences (IRs) [27]. The genome size is approximately 120–160 kb, and differences are mainly caused by the two inverted repeat sequences [28]. The structure and sequence of chloroplast genomes is of great value in revealing the origins and evolution of, and genetic relationship between, species [29]. Firstly, chloroplast genomes are highly conserved and contain rich genetic information, and their structures are simpler than those of nuclear genomes, so it is easier to obtain their full-length sequences. Secondly, the nucleotide substitution rate is moderate (approximately one third the rate of nuclear genes and three times that of the mitochondria) [30]. Moreover, the significantly different molecular evolution rates of the coding and noncoding regions can be applied to the study of different taxonomic categories [31]. At present, in the public database National Center for Biotechnology Information (https://www.ncbi.nlm.nih.gov/, accessed on 15 June 2022), there are chloroplast genome data for the genus *Pourthiaea* that can be queried. However, some recent studies have attempted to solve the phylogenetic relationship between *Pourthiaea* and its relatives, while no studies have been conducted on the chloroplast genome characteristics of the genus.

In this study, the chloroplast genome sequences of ten species of *Pourthiaea* that have been published in GenBank thus far were downloaded, and their characteristics were analyzed comparatively to improve our understanding of the chloroplast genome structure of *Pourthiaea*. The purposes of this study were to compare the genomic structural variation of *Pourthiaea*; to study the contraction and expansion of IRs in the chloroplast genome of *Pourthiaea* and screen hypervariable sites, repeated sequences, and SSRs; and to reveal the relationships among species of *Pourthiaea* and between *Pourthiaea* and related genera through phylogenetic analysis based on plant chloroplast genomes. Our research results

can provide resources for the subsequent interspecific identification, marker development and utilization, genetic breeding, and phylogenetic study of *Pourthiaea*.

## 2. Materials and Methods

### 2.1. Sequence Data Acquisition

The chloroplast genome sequences and annotation information of *P. villosa*, *P. amphidoxa*, *P. pilosicalyx*, *P. blinii*, *P. zhejiangensis*, *P. tomentosa*, *P. hirsuta var. lobulata*, *P. arguta var. salicifolia*, *P. arguta,* and *P. sorbifolia*, as well as the chloroplast genome sequences of 13 species of Photinia, 3 species of Stranvaesia, and 2 outgroups (Table A1), were downloaded from the NCBI database for bioinformatics analysis.

### 2.2. Analysis of the Basic Characteristics of Chloroplast Genomes

According to the chloroplast genome sequences of *Pourthiaea* plants published by the NCBI and their annotations, the chloroplast genome information of the 10 species of *Pourthiaea* was statistically analyzed in Geneious software, including the lengths and GC contents of the whole-genome sequences, the four main divisions (LSC, Ira, SSC, and IRb), and gene annotation results. Moreover, the physical map of the chloroplast genomes of *Pourthiaea* plants was drawn in the DRAW Organelle Genome Maps (OGDRAW)(https://chlorobox.mpimp-golm.mpg.de/OGDraw.html, accessed on 28 June 2022) [32] online drawing tool using the GenBank format file, and the annotated genes of the chloroplast genome were statistically analyzed.

### 2.3. Detection of Repetitive Sequences and SSRs

The simple sequence repeats (SSRs) of chloroplast genomic sequences of the 10 species of *Pourthiaea* were detected using MISA (https://webblast.ipk-gatersleben.de/misa/, accessed on 2 July 2022) [33], and the parameters were set with reference to a study by Wu et al. [34]: the minimum repeat number of mononucleotides, dinucleotides, trinucleotides, tetranucleotides, pentanucleotides, and hexanucleotides were set to 10, 5, 4, 3, 3, and 3, respectively. The palindromic repeats, complement repeats, forward repeats, and reverse repeats in the sequences were calculated using an online program called RE-PUTER (https://bibserv.cebitec.uni-bielefeld.de/REPUTER/manual.html, accessed on 3 July 2022) [35], and the specific parameters were determined according to the default setting [36]: the maximum computed repeats was 50 and the minimum repeat size was 8. Moreover, the Tandem Repeat Finder (http://tandem.bu.edu/trf/trf.html, accessed on 3 July 2022) [37] was used to detect tandem repeats, using the default parameter settings.

### 2.4. Analysis of the IR/SC Boundary Region

The chloroplast genomic sequences of the 10 species of *Pourthiaea* from GenBank were uploaded to IRs cope (https://irscope.shinyapps.io/irapp/, accessed on 10 July 2022) [38] in gb format for visual mapping of the IR/SC boundary region genes of the chloroplast genomes, and the correctness of the IRScope mapping was checked against sequence annotations with Geneious. The contraction and expansion of the IR region were analyzed by comparing the gene types and gene positions of the boundary regions of the 10 species.

### 2.5. Analysis of Genomic Differences

The program mVISTA (http://genome.lbl.gov/vista/mvista/submit.shtml, accessed on 18 July 2022) [39] was used to compare the chloroplast genome sequences of *Pourthiaea*. The chloroplast genome of *P. villosa* was taken as a reference for comparison with the remaining 9 species, and the genomes were aligned with the annotations through visual mapping in mVISTA to identify the differences among the 10 chloroplast genomes.

### 2.6. Analysis of Nucleotide Polymorphisms

The chloroplast genomic sequences of the 10 species of *Pourthiaea* were aligned using MAFTF v7 [40]. Next, the nucleotide polymorphisms (Pis) among genomes were calculated using DnaSP v6 software [41], and the parameters were set to a window length of 600 and a step size of 200 to screen out the loci with higher Pi values as hypervariable regions of the chloroplast genome of *Pourthiaea*. The specific loci were determined according to the annotation results of the chloroplast genome.

### 2.7. Phylogenetic Analysis

Phylogenetic tree reconstruction was carried out using the chloroplast gene sequences of 26 species (including 10 species of *Pourthiaea*, 13 species of *Photinia,* and 3 species of *Stranvaesia*) downloaded from the NCBI, with two species (*Eriobotrya henryi* and *Rhaphiolepis lanceolata*) from the genus *Eriobotrya* and the genus *Rhaphiolepis* functioning as the outgroup. The phylogenetic trees were constructed using the maximum likelihood (ML) and Bayesian inference (BI) methods. After aligning these sequences using MAFFT v7 [40], the alignment results were calibrated in BioEdit software [42]. An ML phylogenetic tree was built in IQ-TREE v1.6.7 software [43] using the best model TVM+F+R2, and 1000 repetitions were run to ensure the stability of the evolutionary tree [44]. A BI phylogenetic tree was built using MrBayes v3.2.6 [45]. JModelTest v2.1.10 [46] was used to select the most appropriate alternative DNA model, and the most appropriate model "TVM+I+G" (freqA = 0.3187, freqC = 0.1813, freqG = 0.1738, freqT = 0.3262, R (a) [AC] = 0.9547, R (b) [AG] = 0.9110, R (c) [AT] = 1.0136, R (d) [CG] = 0.2569, R (e) [CT] = 0.9110, R (f) [GT] = 1.000, p-inv = 0.7650, gamma shape = 0.0300) was used to construct the phylogenetic tree. All phylogenetic analysis results were visualized and adjusted using FigTree (version 1.4.3).

## 3. Results and Analysis

### 3.1. Basic Characteristics of Chloroplast Genomes

The basic characteristics of the chloroplast genomes were analyzed, including chloroplast genome size, gene number, GC content, and other information (Figure 1, Table 1). The results showed that the chloroplasts of the ten species of *Pourthiaea* all had typical circular DNA molecular characteristics. The genomes all had total lengths ranging from 160,159 to 160,401 bp and a conservative tetrad structure, including a large single copy region (88,047–88,359 bp), a small single copy region (19,234–19,338 bp), and a pair of reverse repeat regions (26,341–26,401 bp). The GC contents of the whole chloroplast genomes of the ten species of *Pourthiaea* ranged from 36.5% to 36.6%. The GC contents in the LSC region ranged from 34.1 to 36.6%, the GC contents in the SSC region ranged from 30.2% to 30.4%, and the GC contents in the IR region varied from 42.6% to 42.7%. The GC contents in the IR region were higher than those in the SSC region and LSC region.

**Table 1.** Chloroplast genome characteristics of ten species of *Pourthiaea*.

| Species | Size (bp) | | | | Gene Number | | | | GC Content (%) | | | |
|---|---|---|---|---|---|---|---|---|---|---|---|---|
| | LSC | SSC | IR | Total | CDS | tRNA | rRNA | Total | LSC | SSC | IR | Mean |
| *P. villosa* | 88,307 | 19,306 | 26,394 | 160,401 | 83 | 37 | 8 | 128 | 36.5 | 30.3 | 42.7 | 36.5 |
| *P. amphidoxa* | 88,359 | 19,234 | 26,382 | 160,357 | 83 | 37 | 8 | 128 | 34.1 | 30.4 | 42.6 | 36.5 |
| *P. pilosicalyx* | 88,342 | 19,306 | 26,341 | 160,330 | 84 | 37 | 8 | 129 | 34.4 | 30.3 | 42.7 | 36.6 |
| *P. blinii* | 88,171 | 19,338 | 26,399 | 160,307 | 83 | 38 | 8 | 129 | 34.2 | 30.2 | 42.7 | 36.5 |
| *P. zhejiangensis* | 88,183 | 19,329 | 26,394 | 160,300 | 83 | 37 | 8 | 128 | 36.6 | 30.2 | 42.6 | 36.6 |
| *P. tomentosa* | 88,181 | 19,321 | 26,394 | 160,290 | 83 | 37 | 8 | 128 | 34.1 | 30.3 | 42.6 | 36.5 |
| *P. hirsuta var. lobulata* | 88,127 | 19,308 | 26,394 | 160,223 | 83 | 38 | 8 | 129 | 34.2 | 30.3 | 42.7 | 36.5 |
| *P. arguta var. salicifolia* | 88,124 | 19,287 | 26,394 | 160,199 | 83 | 37 | 8 | 128 | 34.3 | 30.4 | 42.7 | 36.6 |
| *P. arguta* | 88,047 | 19,314 | 26,399 | 160,159 | 83 | 38 | 8 | 129 | 34.2 | 30.3 | 42.7 | 36.5 |
| *P. sorbifolia* | 88,239 | 19,330 | 26,401 | 160,371 | 83 | 37 | 8 | 128 | 34.2 | 30.3 | 42.6 | 36.5 |

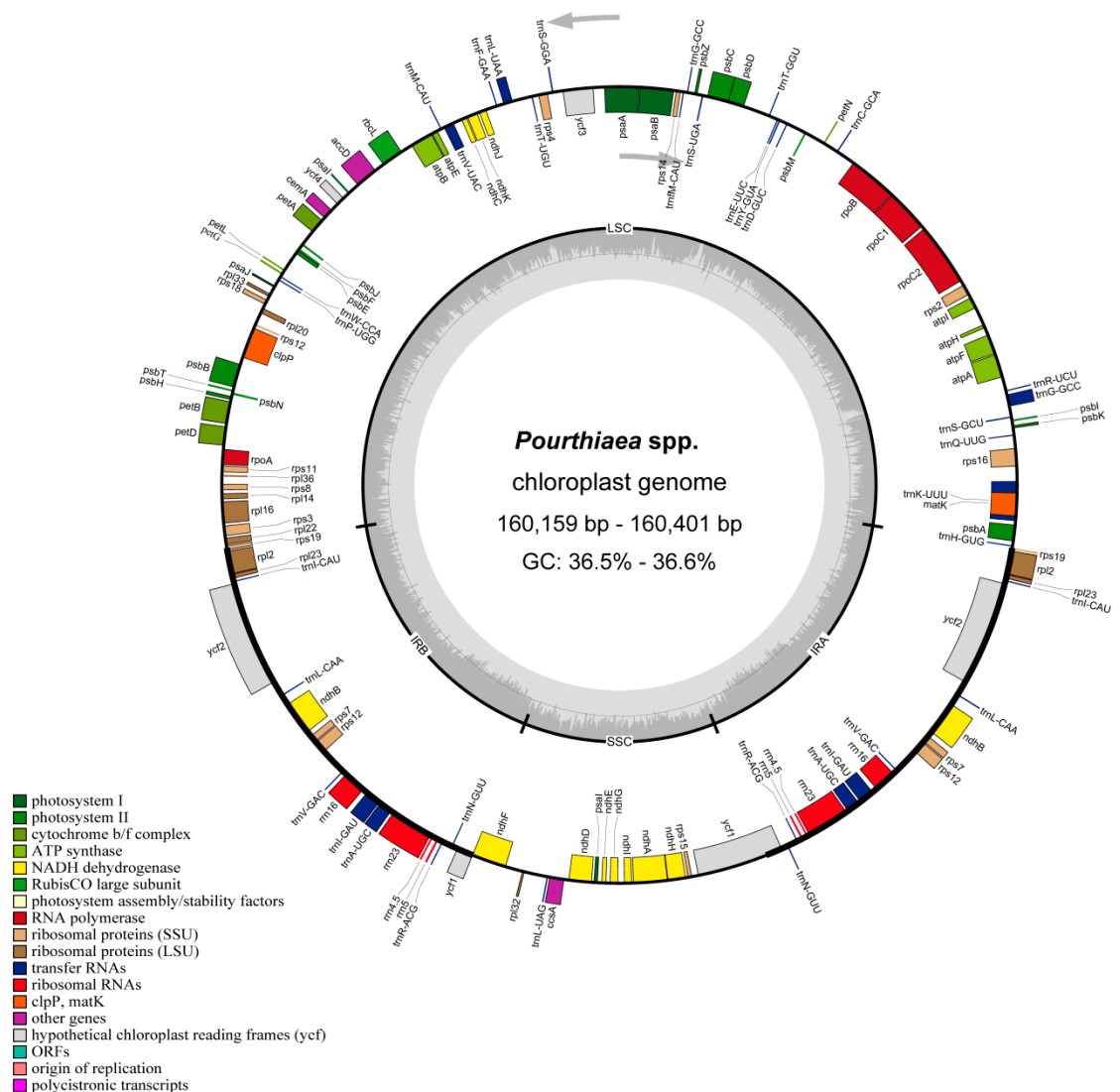

**Figure 1.** Circular map of the chloroplast genome of *Pourthiaea* with annotated genes. Genes shown inside and outside of the circle are transcribed in clockwise and counterclockwise directions, respectively. Genes belonging to different functional groups are color-coded. The GC and AT contents are denoted by the dark gray and light gray colors in the inner circle, respectively.

A total of 130–131 genes were annotated in the chloroplast genomes of the ten species of *Pourthiaea*, including 83–84 protein coding genes. The number of tRNA genes was 37–38, the number of rRNA genes was stable at eight, and there were also two pseudogenes, namely, ψ*ycf1* and ψ*rps19* (Table 2). These genes can be divided into several different categories according to different functions. Among them, 21 genes were located in the IR region. They contained two copies, including nine protein coding genes (*psaI*, *ndhB*, *rps123*, *rps7*, *rpl2*, *rps19*, *rps12*, *ycf1*, and *ycf2*), four rRNA genes (*rrn4.5*, *rrn5*, *rrn16*, and *rrn23*), and eight tRNA genes (*trnA-UGC*, *trnG-GCC*, *trnI-CAU*, *trnI-GAU*, *trnL-CAA*, *trnR-ACG*, *trnN-GUU*, and *trnV-GAC*). Gene transcription regulation is considered to be affected by introns and exons. Introns can accumulate more mutations and play an important role in gene expression regulation. In the chloroplast genome of *Pourthiaea*, 18 genes (12 protein coding genes and six tRNA genes) contained at least one intron, and three genes (*clpP*, *ycf3,* and *rps12*) among them harbored two introns. In addition, *rps12* had a trans-splicing structure, with the 5′ end located in the LSC region and the 3′ end containing an intron located in the IR region.

**Table 2.** List of genes in the chloroplast genomes of *Pourthiaea*.

| Category | Group of Genes | Genes Names | Amount |
|---|---|---|---|
| **Photosynthesis gene** | Photosystems I | *psaA, psaB, psaJ, psaI* (×2) | 5 |
| | Photosystems II | *psbA, psbB, psbC, psbD, psbE, psbF, psbH, psbJ, psbK, psbI, psbM, psbN, psbT, psbZ* | 14 |
| | Cytochrome b/f complex | *petA, petB \*, petD \*, petG, petL, petN* | 6 |
| | ATP synthase | *atpA, atpB, atpE, atpF \*, atpH, atpI* | 6 |
| | NADH dehydrogenase | *ndhA \*, ndhB \* (×2), ndhC, ndhD, ndhE, ndhF, ndhG, ndhH, ndhI, ndhJ, ndhK* | 12 |
| | Rubisco Large subunit | *rbcL* | 1 |
| **Self-replication gene** | RNA polymerase | *rpoA, rpoB, rpoC1 \*, rpoC2* | 4 |
| | Ribosomal proteins (SSU) | *rps2, rps3, rps4, rps7 (×2), rps8, rps11, rps12 \*\* (×2), rps14, rps15, rps16 \*, rps18, rps19 (×2)* | 16 |
| | Ribosomal proteins (LSU) | *rpl2 \* (×2), rpl14, rpl16, rpl20, rpl22, rpl23(×2), rpl32, rpl33, rpl36* | 11 |
| | Transfer RNAs | *trnA-UGC \* (×2), trnC-GCA, trnD-GUC, trnE-UUC, trnF-GAA, trnfM-CAU, trnG-GCC \* (×2), trnH-GUG, trnI-CAU (×2), trnI-GAU \* (×2), trnK-UUU \*, trnL-CAA (×2), trnL-UAA \*, trnL-UAG, trnM-CAU, trnN-GUU (×2), trnP-UGG, trnQ-UUG, trnR-ACG (×2), trnR-UCU, trnS-GCU, trnS-GGA, trnS-UGA, trnT-GGU, trnT-UGU, trnV-GAC (×2), trnV-UAC\*, trnW-CCA, trnY-GUA* | 37 |
| | Ribosomal RNAs | *rrn4.5 (×2), rrn5 (×2), rrn16 (× 2), rrn23 (×2)* | 8 |
| **Other genes** | Maturase | *matK* | 1 |
| | Envelop membrane protein | *cemA* | 1 |
| | Subunit of acetyl-CoA-carboxylase | *accD* | 1 |
| | c-type cytochrome synthesis gene | *ccsA* | 1 |
| | Proteolysis | *clpP \*\** | 1 |
| | Hypothetical chloroplast reading frames (ycf) | *Ycf1 (×2), ycf2 (×2), ycf3 \*\*, ycf4* | 6 |

Note: * Gene contains one intron; ** gene contains two introns; (×2) indicates the number of the repeat unit is 2.

### 3.2. Repetitive Sequences and SSR Analysis

The simple repetitive sequences in the chloroplast genomes of the ten species of *Pourthiaea* were identified using MISA software. A total of 1017 SSR loci were detected, between 95 (*P. arguta* var. *salicifolia*) and 107 (*P. blinii*) loci for each species, including six types of nucleotide repeats, namely mononucleotide, dinucleotide, trinucleotide, tetranucleotide, pentanucleotide, and hexanucleotide repeats, the numbers of which were 762, 181, 3, 60, 10, and 1, respectively (Figure 2A), accounting for 74.93%, 17.80%, 0.29%, 5.90%, 0.98%, and 0.10% of the total SSRs, respectively (Figure 2B). Among them, mononucleotide repeats were the most abundant, followed by dinucleotide repeats, and hexanucleotide repeats were the least abundant, which only existed in *P. villosa*. From the perspective of distribution regions, SSRs were not evenly distributed in various regions in the whole genome. Specifically, SSRs identified in the LSC region were the most abundant (75–83), followed by the SSC region (10–17), while those in the IR region were the least abundant (6–8) (Figure 2C). In the SSRs, there were four types of mononucleotide repeats (A, C, G, and T), three types of dinucleotide repeats (AT, TA, and TC), two types of trinucleotide repeats (AAT and TAA), four types of tetranucleotide repeats (AATA, ATTT, TAAA, and TTTA), four types of pentanucleotide repeats (TCCAA, TGATT, CCTTG, and GGCAA), and one type of hexanucleotide repeat (TAAATA); there were ten motifs in total, among which the dominant motif was A/T (731), followed by AT/AT (171), and AAATAT/ATATTT (1) was the least common, accounting for 71.88%, 16.81%, and 0.10% of the total SSRs, respectively (Figure 2E). Furthermore, 943 tandem repeats, 153 palindromic repeats (CRISPRs), and 347 dispersed repeats (including 236 forward repeats, 104 inverted repeats, and seven com-

plement repeats) were identified in the chloroplast genomes of the ten species of *Pourthiaea*. (Figure 2D).

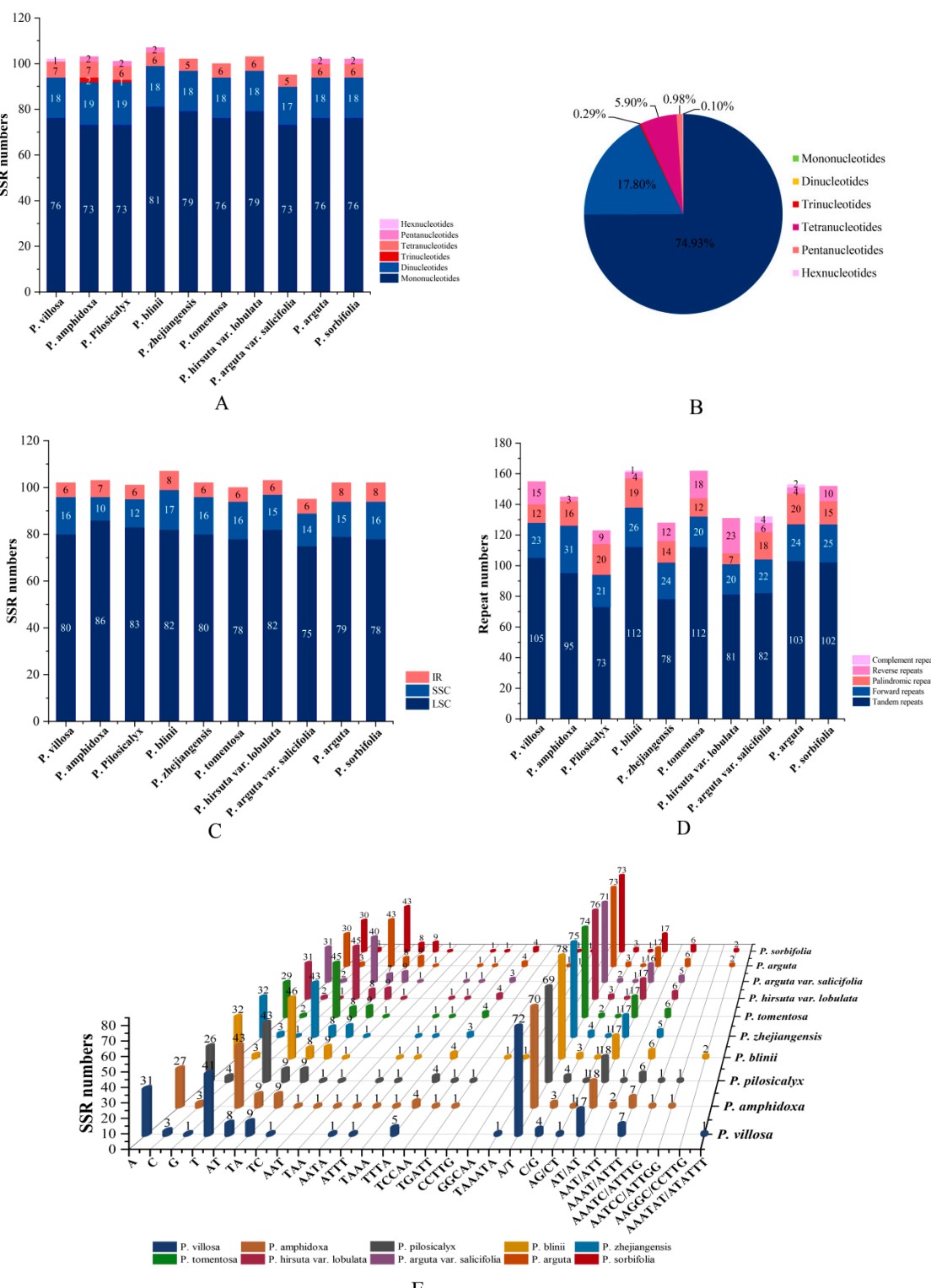

**Figure 2.** Comparison of chloroplast genome repeat sequences in species of *Pourthiaea*. (**A**) Number of different types of SSRs in the chloroplast genomes of ten species of *Pourthiaea*. (**B**) Proportion of different SSR types in the chloroplast genomes of ten species of *Pourthiaea*. (**C**) Number of SSRs in four

chloroplast genome regions of ten species of *Pourthiaea*. (**D**) Number of scattered and tandem repeats in chloroplast genomes of ten species of *Pourthiaea*. (**E**) SSR type information of chloroplast genomes of ten species of *Pourthiaea*.

### 3.3. Analysis of the IR/SC Boundary Region

The analysis of the contraction and expansion of the SC/IR boundary in the chloroplast genomes of *Pourthiaea* (Figure 3) showed that the gene types and gene distribution of the LSC/IRb (JLB), IRb/SSC (JSB), SSC/IRa (JSA), and IRa/LSC (JLA) connection boundaries of the chloroplast genomes of the ten species of *Pourthiaea* varied little. The genes *rps19*, *ycf1*, *ndhF*, and *trnH* were distributed at the LSC/IR and SSC/IR boundaries. The LSC/IRb and IRa/LSC boundaries of the ten species of *Pourthiaea* were all related to the *rps19* gene. The LSC/IRb boundary was located in the gene coding region of *rps19*, and the gene length was 279 bp, of which 159 bp were located in the LSC region and 120 bp expanded to the IRb region. The IRa/LSC boundary was located in the noncoding region between the *rps19* gene and the *trnH* gene, of which the *trnH* gene was 6–150 bp away from the boundary. The *ndhF* gene of *P. arguta* was the closest to the boundary, while that of *P. hirsuta* var. *lobulata* was farthest from the boundary. The IRb/SSC and SSC/IRa boundaries of these ten species of *Pourthiaea* were all related to the *ycf1* gene. The IRb/SSC boundary was located in the overlapping region of the *ycf1* gene and the *ndhF* gene, of which the *ycf1* gene had 7 bp crossing the boundary, located in the SSC region, and the *ndhF* gene had 14 bp crossing the boundary, located in the IRb region. The SSC/IRa boundaries of all ten chloroplast genomes were located in the coding region of the *ycf1* gene, where the length of the *ycf1* gene was 5640 bp, in which 1076 bp expanded into the IRa region. In general, in the ten species of *Pourthiaea*, except for the *trnH* gene located at the IRa/LSC boundary, the genes distributed at the other three boundaries were exactly the same. The above analysis results showed that the SC/IR boundary regions in the chloroplast genomes of the ten species of *Pourthiaea* varied little, and the IR region was highly conserved.

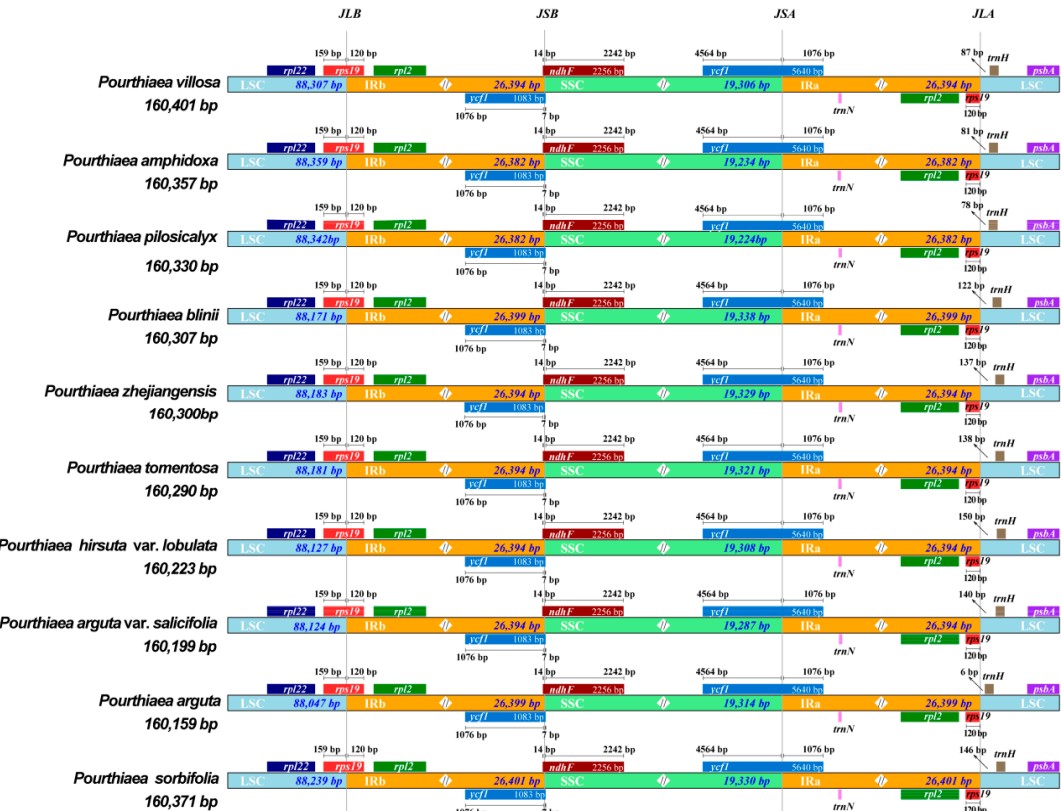

**Figure 3.** Comparison of LSC, SSC, and IR boundary regions of ten species of *Pourthiaea*. The genes

around the edge are displayed above or below the main line. JLB, JSB, JSA, and JLA represent the connection sites of LSC/IRb, IRb/SSC, SSC/Ira, and IRa/LSC, respectively.

### 3.4. Analysis of Genomic Differences

With the chloroplast genome of *P. villosa* as a reference, the chloroplast genomes of nine species of *Pourthiaea*, including *P. amphidoxa*, were aligned and analyzed in full sequences based on the mVISTA software (Figure 4). The results showed that the chloroplast genomes of the ten species of *Pourthiaea* were relatively conserved overall: the IR region was more conserved than the LSC and SSC regions, and the coding region was more conserved than the noncoding region. Variation mainly occurred in the spacer regions of adjacent genes, such as *rpl2-trnH (GUG)*, *trnR (UCU)-atpA*, *trnT (GGU)-psbD*, and *trnT (UGU)-trnL (UAA)*.

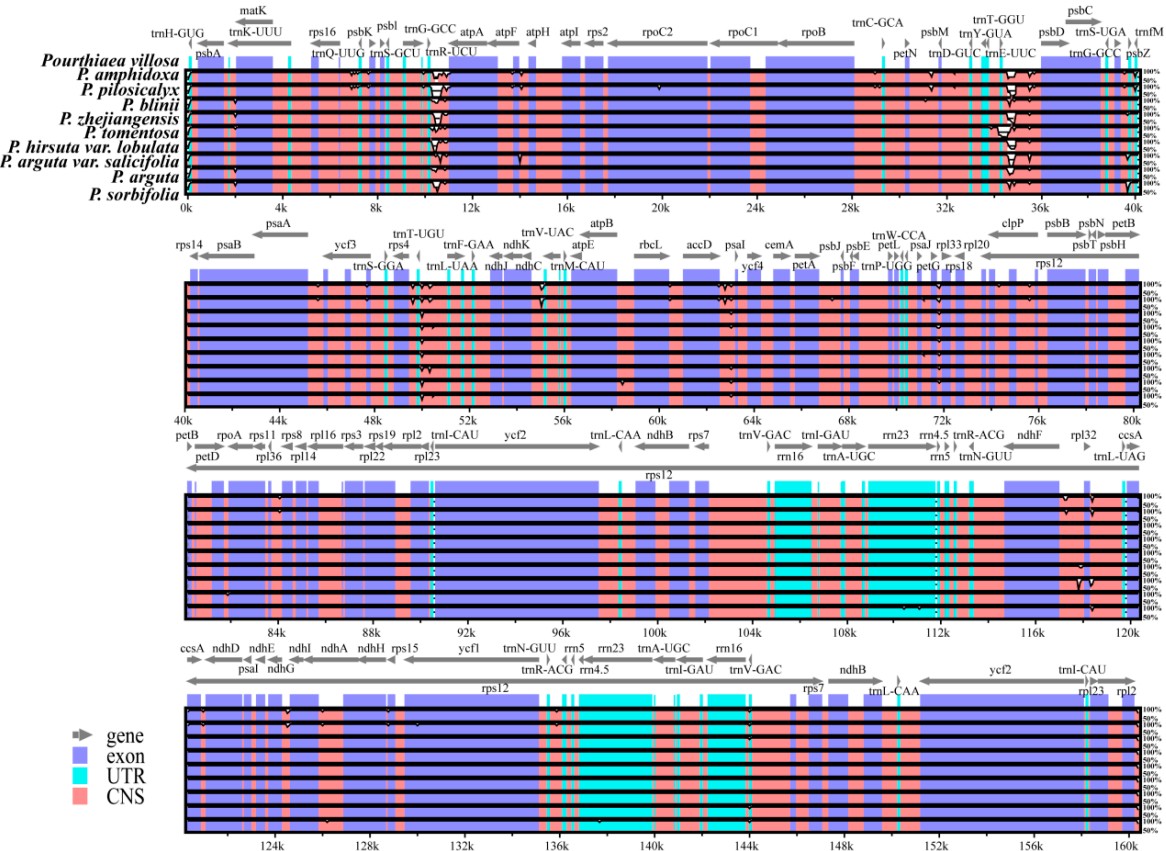

**Figure 4.** Global alignment of the chloroplast genomes of ten species of *Pourthiaea*.

### 3.5. Analysis of Nucleotide Polymorphisms

The nucleotide polymorphisms (Pi values) of the ten chloroplast genome sequences of *Pourthiaea* after MAFFT alignment were calculated in DnaSP v6 software. The results showed that the Pi values in the chloroplast genomes of the ten species of *Pourthiaea* ranged from 0 to 0.00667, with an average of 0.000959. When Pi > 0.006, three hypervariable regions were detected. Among them, two were detected in the LSC region: *trnQ (UUC)-psbk-psbl* (Pi = 0.00604) and *accD-psal* (Pi = 0.00633); and one was detected in the SSC region: *ndhF-rpl32-trnL (UAG)* (Pi = 0.00667) (Figure 5). In general, the Pi values of the ten chloroplast genomes were small, indicating that nucleotides varied little among the ten species of *Pourthiaea*.

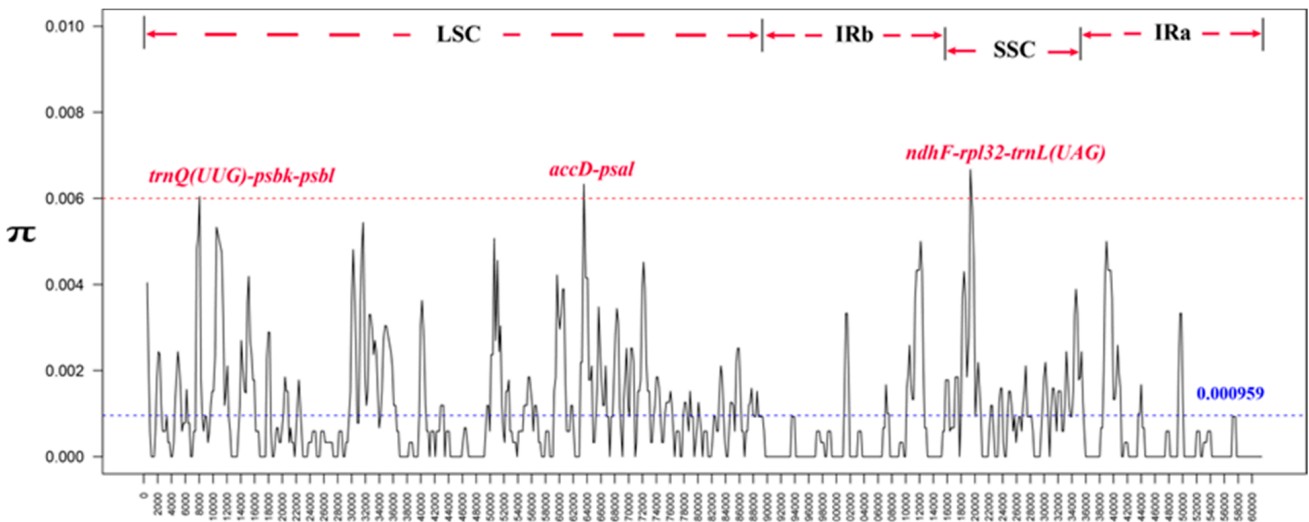

**Figure 5.** Nucleotide polymorphism analysis of chloroplast genomes of ten species *of Pourthiaea*.

### 3.6. Phylogenetic Analysis

With *E. henryi* Nakai and *R. lanceolata* Hu as the outgroups, the phylogenetic relationship between *Pourthiaea* and its related genera (*Photinia* and *Stranvaesia*) was reconstructed based on the complete chloroplast genomes using ML and BI phylogenetic analysis methods (Figures 6 and 7). The results showed that the topological structures of the phylogenetic BI tree and ML tree were basically the same. The bootstrap support (BS) and posterior probability (PP) values of most branches were high, but the branch support values of the BI tree were higher than those of the ML tree. The 26 species of plants were divided into three branches. Thirteen species of *Photinia*, three species of *Stranvaesia* and ten species of *Pourthiaea* were clustered into one branch, while *Photinia* and *Stranvaesia* formed sister groups, and there was no nesting phenomenon among the three genera. The ten species of the genus *Pourthiaea* were divided into two small branches, one branch of which consisted of *P. amphidoxa* and *P. pilosicalyx*, which were sisters to each other. The genetic relationship was well supported (ML-BS = 100%, BI-PP = 1.00). The other eight species, including *P. villosa,* were clustered into one branch, in which *P. hirsuta* var. *lobulata* showed slightly different locations in the ML and BI trees. In the BI tree, *P. hirsuta* var. *lobulata* was separated (BI-PP = 1.00), so it was far from the other seven species. In the ML tree, *P. villosa* and *P. arguta* var. *salicifolia* were separated out, and though the genetic relationship between *P. hirsuta* var. *lobulata* and the other five species was closer, the support of their genetic relationship was not high (ML-BS = 48%). Similarly, the sister relationship of *P. Zhejiangensis* and *P. tomentosa* was not highly supported in the ML tree (ML-BS = 44%), but it was well supported in the BI tree (BI-PP = 1.00).

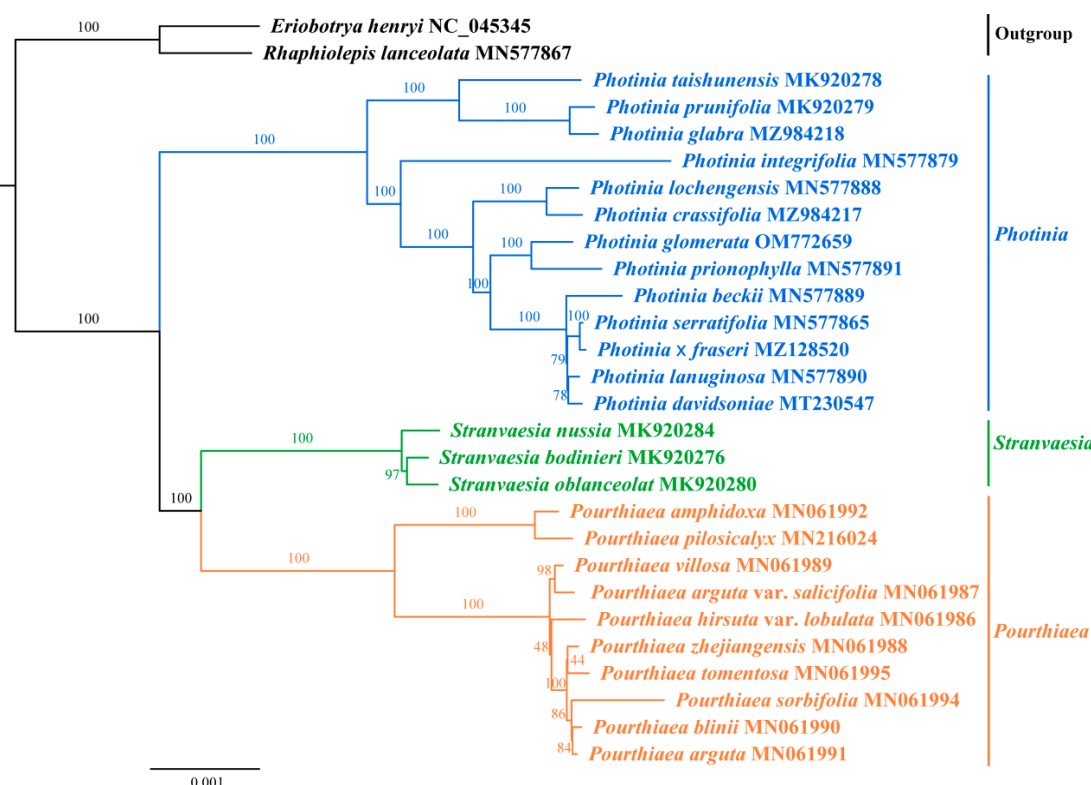

**Figure 6.** ML tree based on the chloroplast genome sequences of 28 species.

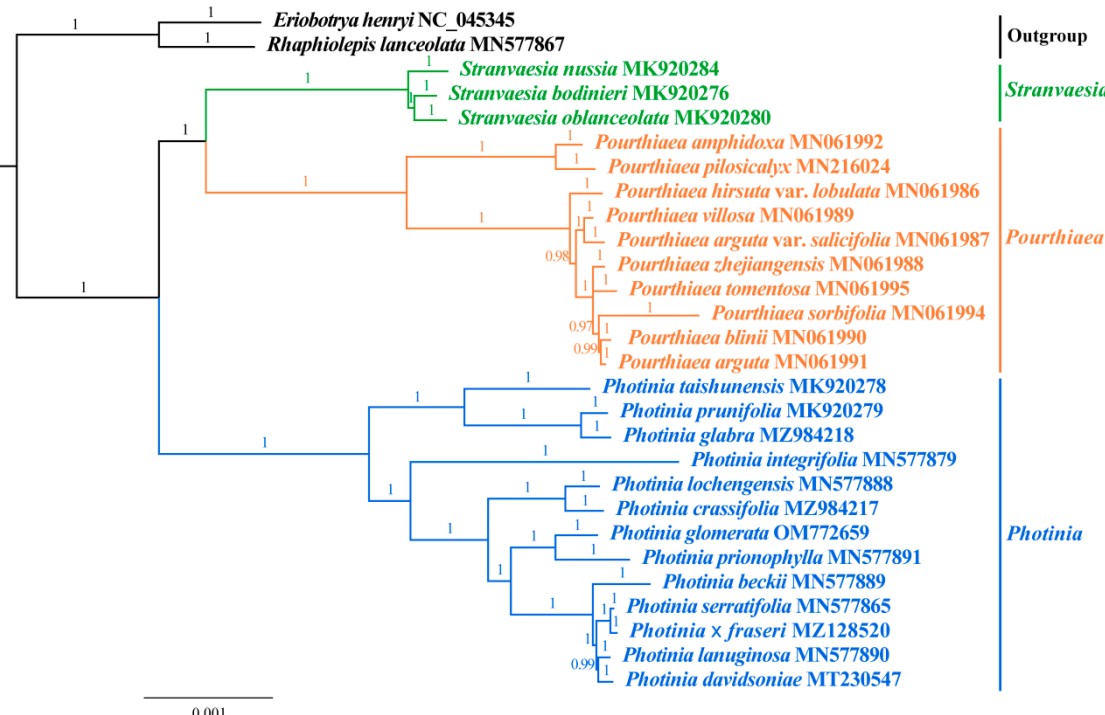

**Figure 7.** BI tree based on the chloroplast genome sequences of 28 species.

## 4. Discussion

The chloroplast genome is a set of DNA sequences carrying genetic information which plays an important role in studying phylogeny, genetic diversity, speciation mechanisms, and so on [47,48]. In this study, through comparative analysis of the chloroplast genomes of

ten species of *Pourthiaea*, it was found that the chloroplast genomes of *Pourthiaea* were all of a traditional tetrad structure with a total length of 160,401–160,159 bp, and 130–131 genes were annotated. Their genome lengths and gene contents are consistent with the characteristics of chloroplast genomes in angiosperms [49]. GC content plays an important role in genome recognition [50]. The GC contents of the chloroplast genomes of the ten species of *Pourthiaea* changed little, ranging from 36.5% to 36.6%, with differences not exceeding 0.1%. GC content was differed across different regions of the chloroplast genomes, among which the IR region had the highest GC content, and the SSC region showed the lowest value. Some studies show that this may be because there are four kinds of rRNA genes with high GC contents distributed in the IR region, while the GC content of the NADH deoxidase gene distributed in the SSC region is very low [51]. GC content affects the stability of sequences. The higher the GC content of a genome, the greater the DNA density, and the more conservative and inflexible the sequence [52]. The higher GC content of the IR region makes it more conserved than the LSC region and the SSC region, which is of great significance for protecting the base contents of chloroplast genomes and stabilizing the genome structures.

Chloroplast genome SSRs are characterized by unilineal inheritance, strong conservatism and simple structure, and they have the advantages of nuclear genome SSR codominance, high polymorphism, and wide distribution, so they are mostly used in population genetics, species evolution, gene flow, and other applications [53,54]. Through SSR analysis of the chloroplast genomes of the ten species of *Pourthiaea*, six SSRs were detected, from mononucleotide to hexanucleotide repeats, among which mononucleotide repeats were the most abundant and hexanucleotide repeats were the least abundant, which is consistent with the results of Shen et al. [55] on thirteen species of *Rosa* in the same family. From the chloroplast genomes of the ten species of *Pourthiaea*, 95–107 SSRs were detected in total, with the fewest in *P. arguta* var. *salicifolia*. Hexanucleotide repeats only existed in *P. villosa*, while trinucleotide repeats were found only in *P. amphidoxa* and *P. pilosicalyx*. In the SSRs from the chloroplast genomes of the ten species of *Pourthiaea*, the dominant repeat motif was A/T, with 731 in total, while there were only 31 G/C, and the poly-A/T type was much larger than the G/C type, which is consistent with the A/T enrichment in the complete chloroplast genome of angiosperms [56]. In this study, 1017 SSRs were identified from the chloroplast genomes of the ten species of *Pourthiaea*, including six types of nucleotide repeats. The repeat motifs were diverse in type, indicating that there was rich SSR polymorphism information in the chloroplast genomes of the species of *Pourthiaea*, which could provide a reference for the identification and genetic diversity analysis of species of *Pourthiaea* [57].

There are four boundaries (JLA, JLB, JSA, and JSB) in chloroplast genomes, which are located in the middle of two IRs and two single copy regions [58], and the contraction and expansion of the IR region can be analyzed by comparing the gene distribution of the four boundary regions. The base substitution rate of the IR region gene is only a quarter of the rate of the SC region, which is of great significance for maintaining the stability of the structures of chloroplast genomes [59]. The contraction and expansion of the IR region of chloroplast genomes is a common phenomenon, which can be observed in plants with close or distant genetic relationships [60,61]. By analyzing the IR boundary of the chloroplast genomes of the ten species of *Pourthiaea*, it was found that the *rps19, ndhF,* and *ycf1* genes crossed the boundary and expanded to the IR region, which is consistent with the research results of Li et al. [62] on the IR/SC boundary of ten species of *Pourthiaea* and five related species. The contraction and expansion of the IR region may cause the production of pseudogenes, gene duplication, and the deletion of single-copy genes [63–65]. In this study, pseudogenes *ycf1* and *rps19* were observed at the IRb/SSC and IRa/LSC boundaries, and these two pseudogenes were also found in *Crataegus* plants of the same family [36]. It has been pointed out that contraction and expansion of the IR region and changes in gene spacer length will lead to changes in chloroplast genome length [66]. By comparing the IR boundary in the chloroplast genomes of the ten species of *Pourthiaea*, it was found

that the genes distributed at the three boundaries (LSC/IRb, Irb/SSC, and SSC/Ira) were completely the same across the ten species of *Pourthiaea*. The difference at the boundary of IRa/LSC was the distance between the *trnH* gene and the boundary ranged from 6 to 150 bp. Therefore, it was speculated that the variation in the chloroplast genome of *Pourthiaea* was mainly caused by the variation in gene spacers. Multiple sequence alignment analysis and nucleotide polymorphism analysis also showed that the variation in gene spacers was much greater than that in protein coding regions. The contraction and expansion of the IR boundary may be helpful in the study of evolutionary models. Compared with more diversified species, species with close relatives will show high similarity at the four connection boundaries of chloroplast genomes [67]. In this study, the four boundaries of the chloroplast genomes of the ten species in the genus *Pourthiaea* were highly conservative. In general, the analysis of the contraction and expansion of the IR region broadens our understanding of the structure and evolution of the chloroplast genome of *Pourthiaea*.

Multiple sequence alignment analysis showed that the sequence variation of the chloroplast genomes of the ten species of *Pourthiaea* was small. The sequence variation in the IR region was smaller than that in the SC region, and the variation in the coding region was smaller than that in the noncoding region, which is consistent with the results of chloroplast genomes of other angiosperms reported [68,69]. Through nucleic acid diversity analysis, it was found that the genome Pi values ranged from 0 to 0.00,667, which was relatively low overall. At Pi > 0.006, three variation sites were screened out, including two in the LSC region and one in the SSC region. There were also two relatively variable sites with Pi values of 0.00333 and 0.005 in the IR region, which is below 0.006, and high nucleic acid variability in the IR region has also been found in studies of 7 *Dracaena* species and 16 other representatives of Asparagales. [70]. It is different from the highly conserved IR region of most plant chloroplast genomes, which may be caused by the differential evolution rate of chloroplast genomes in different regions and species [71]. The hypervariable regions screened here can be used as specific DNA barcode candidate fragments for species identification [72].

Chloroplast genomes play an important role in phylogenetic analysis and have been widely used in the phylogenetic study of angiosperms. Based on the whole chloroplast genome sequences, the phylogenetic analysis of *Pourthiaea* and its two related genera was carried out using the ML and BI methods. The topological structures of the two were basically the same. However, in the ML tree, the support rates of some internal nodes of the *Pourthiaea* branch were low, while the support rates of all branches in the BI tree were high. Therefore, the BI method was selected as the main method used to construct the phylogenetic tree. *Pourthiaea* is closely related to *Photinia*, and its phylogenetic location is controversial. The phylogenetic results of this study showed that *Photinia* and *Pourthiaea* were clustered in different branches, and the clustering between genera was clear, without the nesting phenomenon of species between genera and the monophyly of *Pourthiaea* was highly supported (ML-BS = 100%, BI-PP = 1.00). These results were consistent with the research results of Zhang [6] and Liu [23], supporting the independence of the genus *Pourthiaea*. Wang [22] and Liu [23] transferred *S. amphidoxa* and *S. tomentosa*, which originally belonged to the genus *Stranvaesia* and the genus *Pourthiaea*, but the two were distantly related and located in two different branchlets in the phylogenetic tree based on the complete chloroplast genomes. Specifically, *P. amphidoxa* and *P. pilosicalyx* were sisters, while *P. tomentosa* and *P. zhejiangensis* were sisters. However, it is different from the results of Liu [23], who used nrDNA to reconstruct the phylogenetic relationships within the genus. In the phylogenetic tree based on nrDNA sequences, *P. amphidoxa*, *P. pilosicalyx,* and *P. tomentosa* were closely related and clustered into a single branch, in which *P. pilosicalyx* and *P. tomentosa* were sisters. The phylogenetic relationships based on cpDNA and nrDNA are conflicting. In many studies, the phylogenetic relationships based on nuclear genes and cytoplasmic genes are inconsistent, which may be caused by convergent evolution, incomplete lineage sorting, or hybridization/introgression [73]. Hybridization/introgression may cause chloroplast capture events and coevolution of nuclear genes, thus distorting the

phylogenetic relationships of related species [74]. Liu et al. [75] found strongly supportive but inconsistent nuclear and chloroplast topological structures in the phylogenetic analysis of Maleae and speculated that the cause might be hybridization and/or chloroplast capture events. Furthermore, incomplete lineage sorting and sampling error may also be important reasons for inconsistent phylogenetic relationships. Therefore, a clear definition of species may require further use of genomic data at the population level and more sampling.

## 5. Conclusions

In this study, the characteristics of the chloroplast genomes of the ten species of *Pourthiaea* were compared, and the phylogenetic relationship between *Pourthiaea* and two related genera was inferred based on chloroplast genomes. The results showed that the chloroplast genomes of these ten species of *Pourthiaea* were highly conserved, and there were close similarities among different species in the genus. Meanwhile, three hypervariable regions (*trnQ(UUC)-psbk-psbl*, *accD-psal*, and *ndhF-rpl32-trnL(UAG)*) and 1017 SSR loci were screened out, which, as potential resources for the development of DNA barcodes and effective molecular markers, can be used for further study of the demarcation, phylogeny, population genetics, and evolution of *Pourthiaea*, as well as the molecular breeding and protection of *Pourthiaea* and related genera. Phylogenetic analysis based on complete chloroplast genomes showed that the ten species of *Pourthiaea* were monophyletic, with clear boundaries separating them from *Photinia* and *Stranvaesia*. In summary, this study deepened the understanding of the chloroplast genome structure of *Pourthiaea* and simultaneously provided some basis for further determining the origin and genetic relationships of *Pourthiaea*.

**Author Contributions:** Sequence data acquisition and annotation proofreading, Z.C., W.Z. and F.W.; data analysis, Z.C., Y.X., W.S., Q.L., Y.T., H.Z. and Z.D.; thesis writing, Z.C.; project fund support provider, P.X. All authors have read and agreed to the published version of the manuscript.

**Funding:** Yunnan Science and Technology Talents and Platform Program (No. 202205AF150022).

**Data Availability Statement:** The sequence data used in this paper are all from GenBank.

**Conflicts of Interest:** The authors declare no conflict of interest.

## Appendix A

**Table A1.** GenBank accession number of 28 species.

| Genus | Latin Name | Accession Number |
|---|---|---|
| *Pourthiaea* | *P. villosa* | MN061989 |
| | *P. amphidoxa* | MN061992 |
| | *P. pilosicalyx* | MN216024 |
| | *P. blinii* | MN061990 |
| | *P. zhejiangensis* | MN061988 |
| | *P. tomentosa* | MN061995 |
| | *P. hirsuta* var. *lobulata* | MN061986 |
| | *P. arguta* var. *salicifolia* | MN061987 |
| | *P. arguta* | MN061991 |
| | *P. sorbifolia* | MN061994 |
| *Stranvaesia* | *S. nussia* | MK920284 |
| | *S. bodinieri* | MK920276 |
| | *S. oblanceolata* | MK920280 |

**Table A1.** *Cont.*

| Genus | Latin Name | Accession Number |
|---|---|---|
| Photinia | *Ph. serratifolia* | MN577865 |
| | *Ph. lochengensis* | MN577888 |
| | *Ph. lanuginosa* | MN577890 |
| | *Ph.* × *fraseri* | MZ128520 |
| | *Ph. crassifolia* | MZ984217 |
| | *Ph. prunifolia* | MK920279 |
| | *Ph. glabra* | MZ984218 |
| | *Ph. integrifolia* | MN577879 |
| | *Ph. taishunensis* | MK920278 |
| | *Ph. beckii* | MN577889 |
| | *Ph. glomerata* | OM772659 |
| | *Ph. prionophylla* | MN577891 |
| | *Ph. davidsoniae* | MT230547 |
| *Eriobotrya* | *E. henryi* | NC_045345 |
| *Rhaphiolepis* | *R. lanceolata* | MN577867 |

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
