# Peer review of "Characteristics of the Complete Chloroplast Genome of Pourthiaea (Rosaceae) and Its Comparative Analysis"

_horticulturae, doi:10.3390/horticulturae8121144_

Round 1
Reviewer 1 Report
In this paper, Cao et al, presents a study on the characteristics of the chloroplast genome of Pourthiaea decne and its comparative analysis
Very important comment authors need to urgently correct the name of the plant in the title by following the nomenclature for the scientific names of species
Abstract
The abstract is well-written and summarizes the findings and conclusions of the authors. I think it is too long they can skip some details.
Introduction
The second and third paragraphs can be merged and shortened a bit. In the last paragraph, the authors present the questions in a nice way and why this paper is important.
Materials and Methods
2.1 It is crucial to follow the binomial scientific nomenclature on lines 120 -122
2.2 it will be important to mention whether the default parameters were used for each of the software mentioned in this section as well as in the remaining sections of materials and methods
Results
Results were nicely described no comments.
Discussion
No commnets
Reviewer 2 Report
This manuscript is a good and solid work and represent – as the authors themselves stated – a very good resource for future study on the genus Pourthiaea.
Thus, I have only some formal recommendations:
Line 60: delete the full stop within the sentence after “Photinia”.
Line 96: If you use “Second” you should have a “first”
Line 235: pentanucleotide repeats – the first mentioned in this category is a tetranucleotide one. I guess it should be “TCCAA”?
Line 353-355: I would delete the sentence regarding the identification of species because it is repeated in lines 361-363.
Line 450: Delete the full stop after Pourthiaea
Reviewer 3 Report
I have carefully read the manuscript entitled " Characteristics of the complete chloroplast genome of Pourthiaea Decne. and its comparative analysis” which was submitted for consideration in the Horticulturae (MDPI).
Pourthiaea is a controversial genus of plants in the family (Rosaceae). Its native range is East Asia and Indo-China. It is found in the regions of Assam, Bangladesh, (north-central, south-central and southeastern) China, East Himalaya, Japan, Korea, Laos, Myanmar, Taiwan, Thailand, Tibet and Vietnam. Although the genus is widespread in many regions of the world, its systematic classification has not yet been definitively established. Therefore, I believe that the topic of the article is timely and very important for the scientific community. In this study, to explore the genetic structure and evolutionary characteristics of the chloroplast genome of Pourthiaea, comparative genomics and phylogenetic analysis were conducted using 10 published complete chloroplast genomes from the NCBI database. Phylogenetic analysis showed that the 10 species of Pourthiaea were monophyletic, with clear boundaries with Photinia and Stranvaesia. This is a very important result, ordering the current genus classification system.
This manuscript is in general well written, logically structured, well-illustrated and easy to understand. It also addresses a subject of great interest in the scientific community. The title clearly describes the content of the article, although I propose to change it slightly (I have provided details below). The abstract is well written. The introduction is generally well written as it gives a good background of the research in question. However, I believe that this chapter should be supplemented with data on the research object, please indicate where the genus comes from and what is its natural range. I believe that the Materials and Methods section is well-structured and scientifically sound. The results are well presented, figures and tables are correct. The discussion is very well written, supports the study's conclusion, and skillfully connects its findings to earlier research. My comments mostly relate to relatively minor issues of interpretation and writing. These comments do not influence a positive impression of the article.
Suggestions:
Title: Please consider changing the title to " Characteristics of the complete chloroplast genome of Pourthiaea (Rosaceae) and its comparative analysis"
Lines 12-12, the first sentence of the abstract requires rephrasing
Line 43: … corymbos [x] – reference needed
Line 78-79: “The controversial taxonomic status of the genus makes the published names unstable, and the phenomenon of synonyms is more serious” - sentence requires rephrasing, clarify
Line 120-123- Latin names of taxa should be written in italics
Line 299: Please select accordingly (Photinia Lindl. and Stranvaesia Lindl.) or (Photinia and Stranvaesia)
Line 324-325: “The chloroplast genome is a set of DNA sequences carrying genetic information, which plays an important role in studying phylogeny, genetic diversity, speciation mechanism, etc. [51] - Please add other references
